# Circulating Isthmin-1 Levels and Their Relationship with Diabetes and Metabolic Diseases in Kuwaiti Adults

**DOI:** 10.3390/biomedicines13010101

**Published:** 2025-01-04

**Authors:** Eman Alshawaf, Sulaiman K. Marafie, Mohamed Abu-Farha, Ahmed N. Albatineh, Tahani Alramah, Aldana Albuhairi, Yafa Al Qassar, Reem Zinoun, Rawan Shalabi, Sarah Behbehani, Dalal Mohammed, Fahad Alajmi, Mohammed A. Abdalla, Ebaa Al-Ozairi, Mohammad Shehab, Muhammad Abdul-Ghani, Fahd Al-Mulla, Jehad Abubaker

**Affiliations:** 1Department of Biochemistry and Molecular Biology, Dasman Diabetes Institute, Dasman 15462, Kuwait; eman.alshawaf@dasmaninstitute.org (E.A.); sulaiman.marafie@dasmaninstitute.org (S.K.M.); mohamed.abufarha@dasmaninstitute.org (M.A.-F.); tahani.alramah@dasmaninstitute.org (T.A.); aldanalbuhairi@gmail.com (A.A.); yafa.qassar@gmail.com (Y.A.Q.); reemadnan694@gmail.com (R.Z.); rawan-shalabi@hotmail.com (R.S.); sarah.behbehani@ucdconnect.ie (S.B.); dalalomaralk85@gmail.com (D.M.); 2Department of Translational Research, Dasman Diabetes Institute, Dasman 15462, Kuwait; fahad.alajmi@dasmaninstitute.org (F.A.); mohammed.ahmed@dasmaninstitute.org (M.A.A.); abdulghani@uthscsa.edu (M.A.-G.); 3Department of Community and Population Health, College of Health, Lehigh University, Bethlehem, PA 18015, USA; aha424@lehigh.edu; 4Clinical Research Unit, Dasman Diabetes Institute, Dasman 15462, Kuwait; ebaa.alozairi@dasmaninstitute.org; 5Division of Gastroenterology, Department of Internal Medicine, Mubarak Alkabeer University Hospital, Kuwait University, Kuwait City 31470, Kuwait; mohammad.shehab@dasmaninstitute.org; 6Division of Diabetes, University of Texas Health Science Center, San Antonio, TX 78229, USA

**Keywords:** obesity, pre-diabetes, diabetes, MAFLD, metabolic diseases, Isthmin-1, adipokines

## Abstract

Background/Objectives: Obesity and type 2 diabetes (T2D) are associated with significant alterations in various metabolic biomarkers. Isthmin-1 (Ism1) has recently emerged as a potential marker of metabolic health and was shown in animal studies to associate with metabolic-associated fatty liver disease (MAFLD). In this study, we aimed to investigate the circulatory levels of Ism1 in individuals with obesity compared to non-obese individuals and evaluate their association with insulin resistance, MAFLD, and T2D. The primary outcomes of this study are obesity, insulin resistance, MAFLD, and T2D, while the secondary outcome is hypertension; Methods: This is a cross-sectional study involving 450 participants, who were divided based on their obesity status into people with obesity (n = 182) and those without obesity (n = 265). Circulating Ism1 levels were measured by ELISA and were compared between the groups. Insulin resistance was assessed using the homeostasis model assessment of insulin resistance (HOMA-IR), and fatty liver was evaluated using Fibroscan; Results: Our results showed a significant reduction in circulating Ism1 levels in individuals with obesity (*p*-value = 0.002). Ism1 levels were negatively associated with the odds of T2D, possibly suggesting a protective role. Additionally, individuals with higher CAP scores demonstrated significantly lower Ism1 levels, and the Spearman’s rank correlation revealed a negative association between Ism1 and both CAP scores (r = −0.109, *p*-value = 0.025) and insulin resistance (r = −0.141, *p*-value = 0.004). Logistic regression analysis further supported Ism1 as an independent significant protective factor against obesity-related metabolic dysfunction. This significance persisted after adjusting for several confounders. Furthermore, our ROC results indicate that circulatory Ism1 levels possess significant diagnostic capability for identifying individuals with obesity-related metabolic imbalances with an area under the curve of 0.764 (95% CI = 0.718, 0.811). Finally, the adjusted multinomial analysis suggested that higher levels of Ism1 may play a protective role against pre-diabetes (AOR = 0.88, 95% CI = 0.838, 0.925) and T2D (AOR = 0.87, 95% CI = 0.814, 0.934); Conclusions: This study suggests that reduced Ism1 levels are linked to increased insulin resistance, MAFLD, and T2D in obese individuals. Our findings further corroborate the protective role of Ism1 and highlight its potential utility as a biomarker for monitoring obesity-related metabolic diseases.

## 1. Introduction

Metabolic diseases refer to a broad range of diseases by which the process where the body turns food into energy is disrupted [1]. Obesity is categorized as a chronic illness where individuals have a body mass index (BMI) of over 30 kg/m^2^ with excess adiposity. According to the World Health Organization (WHO), in 2022, one in eight people around the world were obese, a figure which has more than doubled since 1990 [2]. Kuwait ranks among the highest countries in the world for obesity, with a substantial proportion of the adult population classified as overweight or obese. Different organs are impacted by obesity, raising the chance of developing other comorbidities, including type 2 diabetes (T2D), insulin resistance, cardiovascular diseases, and cancer. Additionally, obesity can be influenced by a combination of genetic susceptibility, metabolic pathways, and environmental factors [3].

Insulin resistance is associated with obesity, driven by reduced glucose uptake in fat tissue and muscle and altered hepatic insulin action [4]. Consequently, the combination of increased BMI and distributions of abdominal fat has been shown to enhance the likelihood of developing T2D [5]. T2D is characterized by chronic hyperglycemia emanating from deficiencies in insulin production, insulin action, or both. Management of T2D requires a combination of lifestyle modifications and pharmacological interventions [6]. Obesity, insulin resistance, and T2D are closely linked to the accumulation of excess fat in the liver, leading to the development of fatty liver disease (FDL) [7]. FLD progresses through several stages, starting from simple fatty liver (steatosis) and advancing to metabolic dysfunction-associated fatty liver disease (MAFLD). In MAFLD, fat makes up 5–10% of the liver’s weight, which could potentially further progress to metabolic-associated steatohepatitis (MASH). MASH is characterized by severe inflammation that contributes to fibrosis and liver damage [8,9]. It has also been reported that MASH could further evolve to develop cirrhosis, leading to liver failure or even hepatocellular carcinoma (HCC) [10]. Taken together, the various developmental stages of fatty liver highlight the crucial link between metabolic risk factors, such as obesity, insulin resistance, and T2D, and the progression of FLD itself.

One of the key signaling pathways involved in T2D regulation is the AMP-activated protein kinase (AMPK) pathway [11]. AMPK also plays a regulatory role in glycogen metabolism, promoting its production. Other key players involved in insulin signaling are the phosphoinositide 3-kinase (PI3K) and protein kinase B (Akt) signaling pathways implicated in enhancing insulin sensitivity [12]. Additionally, insulin receptor substrate 1 (IRS-1) and phosphoinositide-dependent protein kinase 1 (PDK1) are essential for the overall regulation of insulin sensitivity and hyperglycemia management in T2D [13,14]. The mammalian target of rapamycin (mTOR) is another crucial regulator of insulin signaling that lies both upstream and downstream of Akt. Once activated, it regulates various cellular functions, including protein biogenesis and cell growth [15]. Taken together, the PI3K/Akt/mTOR pathway is a significant player and therapeutic target for the prevention and treatment of metabolic diseases [16]. Recent studies have demonstrated that Ism1 plays a role in regulating glucose uptake via the PI3K/Akt/mTOR signaling pathway [17,18,19]. Ism1 is an adipokine that has also been implicated in different signaling pathways, including cancer suppression and apoptosis regulation [20,21,22]. Recently, Jiang et al. revealed a dual role of Ism1, where it caused increased adipocyte glucose uptake while suppressing hepatic lipid synthesis in a PI3K/Akt/mTOR-dependent manner [23]. This led to improved hyperglycemia and reduced lipid accumulation in mice. This finding suggests a potential therapeutic role for Ism1 to simultaneously treat T2D and FLD, a promising avenue for future research and clinical applications. Other studies have reported reduced Ism1 levels in obese individuals with T2D compared to their non-diabetic counterparts, indicating a potential link between Ism1 deficiency and the development of T2D [24].

Obesity and T2D are highly prevalent in Kuwait and the Arabian Gulf region [25,26]. This study aimed to estimate the correlation between Ism1 levels and several anthropometric and blood biomarkers in the Kuwait population and investigate its potential association with metabolic disease risk factors such as obesity, insulin resistance, MAFLD, and T2D.

## 2. Materials and Methods

### 2.1. Study Population

The Kuwait Adult Diabetes Epidemiological Multidisciplinary (KADEM) program is a study conducted at the Dasman Diabetes Institute, which was approved by the DDI ethical committee (Study Number RA-2019-030) and registered on clinical trials.gov (NCT06115876, accessed on 23 May 2022). This study was conducted in accordance with the ethical framework of the Helsinki Declaration. All participants signed a consent form before participating in this study. A total of 450 participants who were either non-diabetic or diagnosed with type II diabetes were enrolled in the KADEM. People with type 1 diabetes were excluded from participation. Anthropometric measures were recorded, including age, gender, and BMI, as well as clinical lab tests evaluating lipid profiles, liver function tests, and glucose. BMI was calculated by dividing the weight in kilograms by the square of height in meters, an electronic weighing scale determined weight, and portable inflexible measuring bars were used to measure height (BMI = kg/m^2^), where individuals were classified as non-obese (BMI < 30 kg/m^2^) and obese (BMI ≥ 30 kg/m^2^). Blood pressure and heart rate readings were obtained as the average of three measurements, taken using an Omron HEM-907XL digital sphygmomanometer, with a 5 to 10 min rest interval between each measurement. Participants with malignancies, autoimmune diseases, active infections, endocrine disorders other than diabetes, chronic kidney disease, or those who were pregnant or lactating were excluded.

MAFLD was assessed by a trained specialist using vibration-controlled transient elastography (VCTE) Fibroscan elasticity techniques. To assess clinical variables, participants were asked to fast for at least 10 h overnight for blood samples to be collected for measuring lipid and glycemic profiles, including triglycerides (TG), hemoglobin A1c (HbA1c), fasting plasma glucose (FPG), total cholesterol (TC), high-density lipoprotein (HDL), and low-density lipoprotein (LDL). A Siemens Dimension RXL chemistry analyzer was used for glucose and lipid profiles, while HbA1c levels were determined by the VARIANT™ II Hemoglobin Testing System.

### 2.2. Blood Processing

Blood samples were collected in EDTA tubes and centrifuged at 400× *g* for 10 min at room temperature to separate the plasma. The plasma was centrifuged at 800× *g* for 10 min to obtain a clear supernatant, which was aliquoted into fresh tubes. Plasma samples were stored at −80 °C for future tests. For the ELISA assay, the required plasma samples were thawed at room temperature and centrifuged at 10,000× *g* for 5 min to remove any particles or precipitates. Samples were aliquoted into the appropriate plate layout.

### 2.3. Measurement of Circulating Levels of Isthmin-1 by ELISA

Levels of human Ism1 in plasma were determined using the Isthmin-1 ELISA Kit (Human) (Cat. No. AG-45B-0032-KI01) from Adipogen Life Sciences, Liestal, Switzerland. Plasma samples were diluted 1:4 with 1X sample diluent from the kit, and an ELISA assay was performed following the manufacturer’s protocol. The ELISA plate was read at 450 nm using a plate reader within 30 min of stopping the reaction. Color intensity is directly proportional to Ism1 levels in the samples. The concentration of Ism1 in the samples was extrapolated from the Ism1 standard curve. The intra-assay and inter-assay coefficients of variation (CV%) were less than 10%.

### 2.4. Statistical Analysis

All analyses were performed using SPSS V.29.0 software (IBM Corp., Chicago, IL, USA). Data were coded and checked for any abnormalities. Categorical variables were presented as counts and percentages, while continuous variables were presented as the median (IQR) to represent the center due to skewness. To test for an association between categorical variables, the Pearson chi-square test of independence was implemented if the expected cell counts for 80% of the cells were more than 5; otherwise, the Fisher exact test was implemented. The Mann–Whitney U test was implemented to compare the median of the groups due to the non-normality of the variables. To measure the strength of the linear relationship between two continuous variables and due to the presence of outliers, the Spearman’s rank correlation was implemented. Finally, to model the association between the binary outcome and a set of covariates, multiple logistic regression modeling was implemented, and the receiver operating characteristic (ROC) curve was produced to shed light on the model’s predictive ability. Furthermore, the multiple logistic regression model was checked for significance using the omnibus test and for fitting the data well using the Hosmer and Lemeshow test. All tests were two-tailed, and a significant level was set at 5%.

## 3. Results

### 3.1. Study Population Characteristics

The investigated sample involved 450 participants, and a comprehensive descriptive analysis of the comparison of an array of variables stratified by the primary outcome of obesity is presented in Table 1. The sample was divided based on BMI and included a total of 265 non-obese (BMI < 30 kg/m^2^) and 182 obese participants (BMI ≥ 30 kg/m^2^). The results showed a significant difference in the medians for weight, BMI, waist circumference, hip circumference, waist-to-hip ratio, wrist circumference, systolic blood pressure (SBP), diastolic blood pressure (DBP), FBG, albumin, TG, HDL, homeostasis model assessment of insulin resistance (HOMA-IR), Ism1, controlled attenuation parameter (CAP) score, aspartate aminotransferase (ALT), alanine aminotransferase (ALT), ALT/AST ratio, Fibrosis-4 (FIB-4), platelet, and aspartate aminotransferase-to-platelet ratio index (APRI) (AST/platelet) between individuals with obesity compared to those from the non-obese group (*p*-value < 0.05). Additionally, participants with obesity presented significantly increased insulin resistance reflected by a higher HOMA-IR median and a significant increase in the median of CAP scores, while showing significantly lower Ism1 median levels compared to the non-obese group. Furthermore, obese participants showed significantly higher medians for ALT, ALT/AST ratio, and platelets but significantly lower medians for the FIB-4 index and APRI (AST/platelet), as shown in Table 1.

### 3.2. Levels of Ism1 Correlate with Various Clinical Parameters

To better understand the nature of the relationship between Ism1 and clinical covariates, we applied the Spearman’s rank correlation. Our data revealed a significant negative correlation between Ism1 levels and various parameters, including age, BMI, hip circumference, SBP, FBG, HbA1c, HOMA-IR, and CAP score (Table 2). To further explore the relationship between obesity and various variables including Ism1, we employed univariate analysis (Table 3). Our analysis indicated that Ism1, SBP, TG, HDL cholesterol, HOMA-IR, and CAP scores are significantly and independently associated with obesity. Additionally, adjusted analysis demonstrated that increased Ism1 and HDL levels have a protective effect against obesity. Specifically, a one-unit increase in the Ism1 level was associated with a 5.6% decrease in the odds of being obese (AOR = 0.944; 95% CI: 0.904–0.987, *p*-value = 0.010). Similarly, a one-unit increase in HDL is associated with a 47.6% reduction in the odds of being obese (AOR = 0.524; 95% CI: 0.314–0.875, *p*-value = 0.014). It is worth noting that the multiple logistic regression model presented in Table 3 produced an area under the ROC curve of 0.764 (95% CI: 0.718, 0.811) (Figure 1), indicating a good predictive ability for Ism1 to discriminate between obese and non-obese participants. According to the omnibus test, the multiple logistic regression model was significant (chi-square = 92.15, *p*-value < 0.001) and fits the data (chi-square =11.92, *p*-value = 0.155) according to the Hosmer and Lemeshow test.

To investigate the significance of Ism1 further, we examined the relationship between T2D and Ism1 with other variables (Table 4). Our analysis indicated that Ism1 and TG are significantly and independently associated with pre-diabetes and T2D. Adjusted analysis showed that increased Ism1 levels have a protective role against T2D. Compared to people with no diabetes (reference group), a one-unit increase in Ism1 resulted in a 12% decrease in the pre-diabetes odds after adjusting for all other covariates included in the model. Additionally, compared to the reference group, people with no diabetes showed that a one-unit increase in Ism1 resulted in a 12.8% decrease in the odds of having T2D after adjusting for all other covariates included in the model. Our analysis implies that higher levels of Ism1 could potentially provide a protective effect against pre-diabetes or T2D compared to people without diabetes.

## 4. Discussion

Ism1 is a newly studied adipokine with potential roles in obesity and T2D, in particular [23,24]. However, a comprehensive understanding of its distribution, function, and association with various phenotypic parameters remains elusive. Previous studies reported a link between higher levels of Ism1 and improved glucose tolerance with lower T2D risk [23,24], while others linked the severity of albuminuria to a substantial increase in serum Ism1 in people with T2D [24,27]. The primary cause of such differences and ambiguous correlations might have resulted from differences related to sample size, study design, and population characteristics, in particular an incomplete adjustment for potential confounders. In this study, we investigated the circulating Ism1 as a potential biomarker for conditions of metabolic diseases in a well-phenotyped population of individuals with obesity compared to those without obesity. Our data showed a significant reduction in Ism1 levels in people with obesity, in addition to increases in insulin resistance and MAFLD indicated by increased HOMA-IR and CAP scores, respectively. Our analysis also revealed a negative association between Ism1 and odds of T2D, suggesting a protective role for Ism1 against T2D. Similarly, individuals with high CAP scores showed significantly reduced Ism1 levels, and a negative correlation with Ism1 according to Spearman’s rank correlation. As a result, our logistic regression analysis suggests Ism1 as an independent protective factor to monitor obesity progression.

People with obesity, in particular abdominal obesity, are at increased risk of developing T2D and MAFLD [28,29,30]. Several studies reported that in over 50% of patients, T2D was accompanied by MAFLD [31,32]. Many studies have shown that metabolic comorbidities, including obesity, T2D, dyslipidemia, and hypertension, significantly influenced the prevalence and incidence of MAFLD [33]. In line with other studies, our study sample demonstrated that obese individuals, with a median BMI of 33.8 kg/m^2^ and waist size of 109 cm depicting abdominal obesity, were more likely to have increased CAP scores, which is reflective of MAFLD and increased insulin resistance. This group of participants was presented with significantly reduced levels of circulating Ism1 (3.15 ng/mL) compared to those without obesity (5.17 ng/mL). These findings were consistent with those of a recent study by Wang et al., where they reported low levels of Ism1 as an independent risk factor for the development of T2D [24]. However, they did not find a clear link between changes in Ism1 levels and the development of MAFLD in people with T2D [24].

Considering the strong association between Ism1 and obesity, we evaluated the diagnostic potential of circulating Ism1 for obesity using ROC analysis. Our ROC results indicate that serum Ism1 levels possess significant diagnostic capability for identifying individuals with obesity. The area under the ROC curve was 0.764, indicating good predictive ability in distinguishing the outcome. These findings suggest that Ism1 may serve as a promising novel biomarker for assessing obesity-related metabolic disorders and predicting the risk of developing obesity-related comorbidities.

Additionally, our findings indicate that low levels of Ism1 are significantly associated with an increased risk of developing pre-diabetes and T2D. Specifically, compared to people with normal glucose, the adjusted odds ratios suggest that for each unit increase in Ism1, the odds of pre-diabetes decrease by 12% (AOR = 0.88, 95% CI = 0.838, 0.925, *p* < 0.001) and the odds of T2D decreases by 13% (AOR = 0.87, 95% CI = 0.814, 0.934, *p* < 0.001). These results underscore the protective role of Ism1 against the development of diabetes. The significant association implies that maintaining higher levels of Ism1 may be beneficial in reducing the risk of transitioning from normal glucose regulation to pre-diabetes and subsequently to T2D. Given that pre-diabetes is a critical stage where intervention can prevent the progression to T2D, Ism1 could potentially serve as a valuable biomarker for the early identification of individuals at risk. This highlights the importance of further research into Ism1’s mechanisms and its potential as a target for therapeutic strategies aimed at preventing diabetes onset.

Some of this study’s strengths are the large sample size and the rigorous data analysis and biostatistical modeling techniques implemented. Additionally, the covariates used in the analysis were measured in the laboratory, so there was no recall bias. However, some limitations include the fact that, due to the cross-sectional design, the results can be used only to establish associations but not causations or temporality. Also, the sampling conducted was a non-probability sampling technique.

## 5. Conclusions

In conclusion, our findings demonstrated an association between T2D and increased insulin resistance with reduced Ism1 levels. Furthermore, we observed reduced levels of circulating Ism1 in people with obesity and MAFLD. Collectively, our findings suggested a potential protective role for Ism1 against obesity and its complications, indicating it as a potential biomarker or diagnostic tool for early detection of metabolic diseases, warranting further investigation into its therapeutic potential.

## Figures and Tables

**Figure 1 biomedicines-13-00101-f001:**
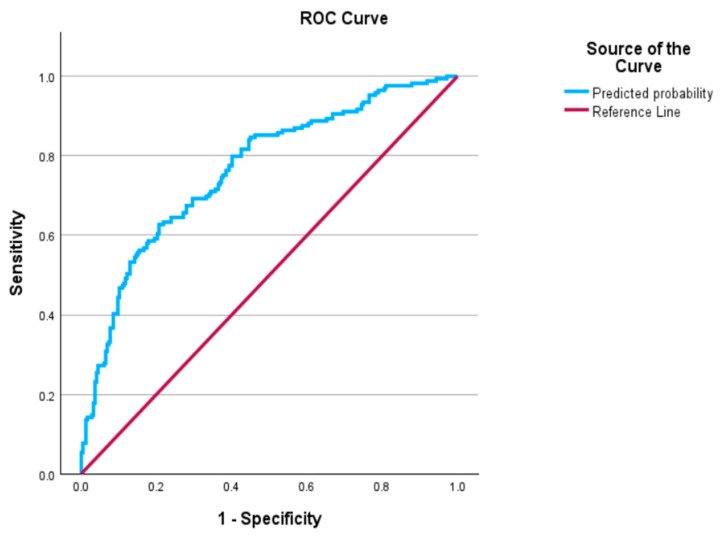
Area under the receiver operating characteristic curve for the model presented in Table 3.

**Table 1 biomedicines-13-00101-t001:** Descriptive analysis of biochemical parameters comparing non-obese with obese participants.

	All Participants n ^c^ = 450 (100%)	Non-Obese (BMI < 30) n = 265 (59.3%)	Obese (BMI ≥ 30) n = 182 (40.7%)	*p*-Value
Gender				0.151 ^a^
Male	234	145 (62.5)	87 (37.5)	
Female	216	120 (55.8)	95 (44.2)	
Age (years)	49 (15.3)	48 (16)	49.5 (14.3)	0.099 ^b^
Height (cm)	167 (15)	168 (15)	165 (15.6)	0.151 ^b^
Weight (kg)	81.7 (25.0)	72 (18)	97 (19.8)	**<0.001**
BMI (kg/m^2^)	28.7 (7.7)	25.9 (4.38)	33.8 (5.15)	<0.001
Waist Circumference (cm)	98 (20.5)	89 (15)	109 (13)	**<0.001**
Hip Circumference (cm)	107 (16)	101 (10)	117 (12.5)	**<0.001**
Waist/Hip Ratio	0.905 (0.11)	0.891 (0.12)	0.931 (0.12)	**<0.001**
Wrist Circumference	17 (1.9)	16.4 (1.85)	17 (1.55)	**<0.001** ^b^
SBP (mmHg)	126 (19)	125 (20)	129 (16)	**<0.001** ^b^
DBP (mmHg)	79 (14)	78 (14)	80 (11)	**0.047** ^b^
Heart Rate	76 (14)	74.5 (14)	78 (13)	0.053 ^b^
FBG (mmol/L)	5.2 (1.1)	5.2 (1.1)	5.3 (0.97)	**0.032** ^b^
Albumin (g/L)	40 (4)	41 (5)	39 (4)	**<0.001** ^b^
Total Chol (mmol/L)	5.15 (1.4)	5.1 (1.5)	5.2 (1.3)	0.402 ^b^
TG (mmol/L)	0.98 (0.71)	0.83 (0.68)	1.14 (0.77)	**<0.001** ^b^
HDL Cholesterol (mmol/L)	1.43 (0.47)	1.46 (0.5)	1.38 (0.41)	**0.026** ^b^
LDL Cholesterol (mmol/L)	3.2 (1.3)	3.2 (1.3)	3.2 (1.3)	0.369 ^b^
HbA1c (%)	5.6 (0.8)	5.5 (0.8)	5.6 (0.9)	0.066 ^b^
HOMA-IR	2.87 (3.2)	2.19 (2.71)	3.83 (3.98)	**<0.001** ^b^
Ism1 (ng/mL)	4.11 (7.45)	5.17 (8.28)	3.15 (5.79)	**0.002** ^b^
CAP Score	247 (74)	232.0 (62)	275.0 (77)	**0.002** ^b^
ALT	28.0 (16)	27 (15)	30 (16)	**0.036** ^b^
AST	20.0 (8)	20 (8)	20 (7)	0.192 ^b^
ALT/AST Ratio	1.44 (0.61)	1.375 (0.61)	1.556 (0.62)	**<0.001 ^b^**
Platelet	266 (83)	260 (84)	273.5 (93)	**0.012 ^b^**
FIB-4 Index	0.65 (0.43)	0.760 (0.35)	0.570 (0.32)	**0.034 ^b^**
APRI (AST/platelet)	0.075 (0.04)	0.078 (0.05)	0.071 (0.04)	**0.005 ^b^**

^a^ *p*-value calculated using Pearson chi-square test of independence, ^b^ *p*-value calculated using Mann–Whitney U test due to non-normality of at least one of the groups. ^c^ The numbers do not add up to N due to missing values. Bold values in the table indicate statistically significant findings (*p* < 0.05).

**Table 2 biomedicines-13-00101-t002:** Spearman’s rank correlation between Ism1 and various parameters in the study participants (n = 450).

Variables	^a^ Correlation	*p*-Value
Age (years)	−0.153	**0.002**
Height (cm)	0.045	0.354
Weight (kg)	−0.093	0.055
Body Mass Index (kg/m^2^)	−0.143	**0.003**
Waist Circumference (cm)	−0.091	0.117
Hip Circumference (cm)	−0.123	**0.011**
Waist/Hip Ratio	0.001	0.999
Wrist Circumference	−0.009	0.850
SBP (mmHg)	−0.138	**0.004**
DBP (mmHg)	−0.023	0.644
Heart Rate	−0.015	0.753
Fasting Glucose (mmol/L)	−0.140	**0.004**
Albumin (g/L)	−0.013	0.785
Total Cholesterol (mmol/L)	0.039	0.426
TG (mmol/L)	−0.080	0.100
HDL Cholesterol (mmol/L)	0.092	0.060
LDL Cholesterol (mmol/L)	0.034	0.493
HbA1c (%)	−0.247	**<0.001**
HOMA-IR	−0.141	**0.004**
CAP Score	−0.109	**0.025**
ALT	0.006	0.900
AST	0.070	0.152
ALT/AST Ratio	−0.057	0.247
FIB-4 Index	−0.155	0.241
Platelet	−0.018	0.713
APRI (AST/platelet)	0.076	0.117

^a^ Due to the skewness of most covariates, the Spearman’s correlation coefficient was implemented to down-weight the effect of outliers. Bold values in the table indicate statistically significant findings (*p* < 0.05).

**Table 3 biomedicines-13-00101-t003:** Multiple logistic regression modeling showing the association between obesity and Ism1 (n = 450).

Covariate	Univariate Analysis OR (95% CI)	*p*-Value	Adjusted Analysis AOR (95% CI)	*p*-Value
Ism1 (ng/mL)	0.937 (0.902, 0.974)	**<0.001**	0.944 (0.904, 0.987)	**0.010**
SBP (mmHg)	1.020 (1.007, 1.033)	**0.002**	1.008 (0.992, 1.025)	0.308
FBG (mmol/L)	1.032 (0.925, 1.152)	0.574	0.788 (0.611, 1.018)	0.068
Total Chol (mmol/L)	1.043 (0.961, 1.131)	0.317	1.026 (0.904, 1.164)	0.696
TG (mmol/L)	1.439 (1.113, 1.860)	**0.006**	1.132 (0.770, 1.663)	0.528
HDL Chol (mmol/L)	0.524 (0.314, 0.875)	**0.014**	0.685 (0.338, 1.386)	0.293
LDL Chol (mmol/L)	1.144 (0.926, 1.413)	0.213	1.015 (0.758, 1.360)	0.920
HbA1c (%)	1.046 (0.932, 1.174)	0.442	0.986 (0.727, 1.338)	0.929
HOMA-IR	1.131 (1.069, 1.197)	**<0.001**	1.076 (1.001, 1.158)	**0.048**
CAP Score	1.017 (1.013, 1.022)	**<0.001**	1.015 (1.010, 1.021)	**<0.001**

Estimates in the adjusted analysis are adjusted for age and gender. OR: odds ratio, AOR: adjusted odds ratio. Bold values in the table indicate statistically significant findings (*p* < 0.05).

**Table 4 biomedicines-13-00101-t004:** Multinomial logistic regression modeling showing the association between diabetes status and Ism1.

Covariate	Pre-Diabetes * AOR (95% CI)	*p*-Value	T2D * AOR (95% CI)	*p*-Value
Ism1 (ng/mL)	0.880 (0.838, 0.925)	**<0.001**	0.872 (0.814, 0.934)	**<0.001**
SBP (mmHg)	1.011 (0.993, 1.029)	0.249	1.014 (0.991, 1.036)	0.229
Total Cholesterol (mmol/L)	1.180 (0.607, 2.295)	0.625	0.987 (0.352, 2.771)	0.980
TG (mmol/L)	1.958 (1.113, 3.444)	**0.020**	2.934 (1.437, 5.992)	**0.003**
HDL Cholesterol (mmol/L)	0.398 (0.145, 1.094)	0.074	0.378 (0.090, 1.588)	0.184
LDL Cholesterol (mmol/L)	0.787 (0.377, 1.642)	0.523	0.648 (0.215, 1.955)	0.441

* Estimates in the adjusted analysis are adjusted for age and gender. AOR: adjusted odds ratio, the outcome reference group is the non-diabetic one. Bold values in the table indicate statistically significant findings (*p* < 0.05).

## Data Availability

The original contributions presented in the study are included in the article, further inquiries can be directed to the corresponding author.

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
