# Peer review of "Circulating Isthmin-1 Levels and Their Relationship with Diabetes and Metabolic Diseases in Kuwaiti Adults"

_biomedicines, 2025, doi:10.3390/biomedicines13010101_

Round 1

Reviewer 1 Report (Previous Reviewer 1)

Comments and Suggestions for Authors

The authors provided the results from Kuwait Adult Diabetes Epidemiological Multidisciplinary (KADEM) Program study conducted at Dasman Diabetes Institute on 450 participants. They aimed to investigate the circulatory
levels of Ism1 in individuals with obesity compared to non-obese individuals and evaluate its association with insulin resistance, fatty liver, and T2D.

Some parts are not clear so these must be clarified before any further consideration:

Abstract: the aim of this study is not coherent with the aim written in the introduction part: lines 23-25 vs. lines 99-101. Are you investigating fatty liver of MAFLD? The authors investigated all different kind of data. Please point the primary outcomes and secondary outcomes.

Introduction:

lines 53-54. Please provide information about the obesity in your Country to compare, not only the whole world, since you are investigating.

line 99: the correlation between the Ism1 and what exactly? The association of what with what? The sentences are not properly written. Please rephrase.

The last sentence in lines 101-103 is unnecessary.

Methods:

should give more detailed than proposed. What are exclusions? Cancers, autoimmunity, infections, endocrine diseases, etc?

line 112: what kind of participants? Make it clear.

Please divide here in different subheadings: primary measures and secondary measures. The blood pressure performing is not explained (SBP, DBP and heart rate).

Please explain all the abbreviations in main text: AST, ALT, SBP, DBP, HOMA-IR, and all others (please check carefully) , not only in the figure legends and abstract.

Results and Discussion: what are limitations of the study?

English needs grammar check.

Comments on the Quality of English Language

moderate

Author Response

Reply to the comments raised by reviewer 1 on the manuscript titled “Circulating Isthmin-1 Levels and Their Relationship with Diabetes and Metabolic Diseases in Kuwaiti Adults” which is submitted for possible publication in the Biomedicine Journal.

The authors thank the reviewer for the comments and suggestions that were raised. Please find below the reply to these comments and suggestions in the same order as appeared in your report. 

Reviewer: Abstract: the aim of this study is not coherent with the aim written in the introduction part: lines 23-25 vs. lines 99-101. Are you investigating fatty liver of MAFLD? The authors investigated all different kind of data. Please point the primary outcomes and secondary outcomes.

Reply: Thank you for the note. Please note that “Metabolic dysfunction-associated fatty liver disease (MAFLD) is the term suggested in 2020 to refer to fatty liver disease related to systemic metabolic dysregulation. The name change from nonalcoholic fatty liver disease (NAFLD) to MAFLD comes with a simple set of criteria to enable easy diagnosis at the bedside for the general medical community, including primary care physicians”. Therefore, to remove any confusion, the term “fatty liver” is replaced by “MAFLD” in the abstract lines 30-31. Also, please note that to conduct a comprehensive analysis, several things have been investigated. For example, the Isthmin-1 level was compared between obese and non-obese subjects and was correlated with other covariates. But when a model was constructed, the Isthmin-1 level was the exposure and both obese vs. nonobese and diabetic vs. non-diabetic were the primary outcomes. It is customary in Epidemiologic research to have the exposures on the rows and the outcome on the columns as we did in Tables 3 and 4. We added the following sentence to the abstract in lines 32-33:
“The primary outcomes of the study are obesity, insulin resistance, MAFLD, and T2D, while the secondary outcome is hypertension.”

Reviewer: Introduction: lines 53-54. Please provide information about the obesity in your Country to compare, not only the whole world, since you are investigating.

Reply: We thank the reviewer for this suggestion. In addition to the sentence on line 106 “Obesity and T2D are highly prevalent in Kuwait and the Arabian Gulf region [25, 26]”, we have added the following sentence (lines 60 – 61) to the introduction to reflect the status of obesity in Kuwait. “Kuwait ranks among the highest countries in the world for obesity with a substantial proportion of the adult population classified as overweight or obese”.

Reviewer: line 99: the correlation between the Ism1 and what exactly? The association of what with what? The sentences are not properly written. Please rephrase.

Reply: Thank you, we have added the sentence “correlation between Ism1 levels and several anthropometric and blood biomarkers in Kuwait population”, see lines (107 – 108).

Reviewer: The last sentence in lines 101-103 is unnecessary.

Reply: As suggested by the reviewer, the sentence lines (101 – 103) removed.

Reviewer: Methods: should give more detailed than proposed. What are exclusions? Cancers, autoimmunity, infections, endocrine diseases, etc?

Reply: Participants with malignancies, autoimmune diseases, active infections, endocrine disorders other than diabetes, chronic kidney disease, or those who were pregnant or lactating were excluded.
The following sentence was added to lines 128-128of page 3:
“Participants with malignancies, autoimmune diseases, active infections, endocrine disorders other than diabetes, chronic kidney disease, or those who were pregnant or lactating were excluded.”

Reviewer: line 112: what kind of participants? Make it clear.

Reply: Participants are the study subjects who were referred to Dasman Diabetes Institute for medical care. A detailed description is available under the “Study Population” subsection and the following sentence was added “A total of 450 participants who were either non-diabetic or diagnosed with type II diabetes were enrolled” for clarity.    

Reviewer: Please divide here in different subheadings: primary measures and secondary measures. The blood pressure performing is not explained (SBP, DBP and heart rate).

Reply: We appreciate the reviewer’s suggestion. Blood pressure and heart rate readings were obtained as the average of three measurements, taken using an Omron HEM-907XL digital sphygmomanometer, with a 5 to 10-minute rest interval between each measurement.
The following sentence was added to lines 128-130 of page 3:
“Blood pressure and heart rate readings were obtained as the average of three measurements, taken using an Omron HEM-907XL digital sphygmomanometer, with a 5 to 10-minute rest interval between each measurement.”

Reviewer: Please explain all the abbreviations in main text: AST, ALT, SBP, DBP, HOMA-IR, and all others (please check carefully), not only in the figure legends and abstract.

Reply: Thank you, all abbreviations are now listed as you suggested, see subsection 2.1 and 3.1 for details. 

Reviewer: Results and Discussion: what are the limitations of the study?

Reply: One paragraph summarizing strengths and limitations is listed right before section 5 (Conclusions). 
“Some of this study's strengths are the large sample size and the rigorous data analysis and biostatistical modeling techniques implemented.  Additionally, the covariates used in the analysis we measured in the laboratory, so there was nothing like recall bias. Some limitations however, due to the cross-sectional design, the results can be used only to establish associations but not causations or temporality. Also, the sampling conducted was a non-probability sampling technique. We hope you are OK with that.”

Finally, please note that the manuscript has gone through some editing by an English native speaker, and we hope that it is OK now.

Reviewer 2 Report (Previous Reviewer 2)

Comments and Suggestions for Authors

The authors have responded to my comments in a satisfactory manner. 

Author Response

We sincerely thank the reviewer for accepting our response.

Round 2

Reviewer 1 Report (Previous Reviewer 1)

Comments and Suggestions for Authors

fine

This manuscript is a resubmission of an earlier submission. The following is a list of the peer review reports and author responses from that submission.

Round 1

Reviewer 1 Report

Comments and Suggestions for Authors

The authors aimed to investigate the circulatory levels of Ism1 in individuals with obesity compared to non-obese individuals and evaluate its association with insulin resistance, fatty liver, and T2D.

Abstract, Methods, Results, Discussion, Conclusion: abbreviations are not fully explained. My biggest concern is the assessment of fatty liver and liver function without involving additional diagnostics as abdominal ultrasound, liver enzymes, coagulation tests and furthermore making statements about fatty liver and liver function.

This raises a question about the validity and reliability of the study results.    

Comments on the Quality of English Language

moderate

Reviewer 2 Report

Comments and Suggestions for Authors

The presented manuscript is an investigation of the relationship of Isthmin-1 levels with obesity, diabetes and some parameters related to these diseases in Kuwait population. The study design is simple but effective. They recruited study subjects, measured some biochemical and anthropometric parameters, Isthmin-1 adipokine was measured with an ELISA, and then performed uni- and multivariate statistical analysis. 

The results obtained are interesting, the methods are adequately described, the introduction is correct and the results support the conclusion. However, there are some points to consider.

1. Although Figure 1 is interesting, it does not really have a clinical application, because it is easier and cheaper to diagnose obesity through BMI.

2. In Table 2, the authors show some significant but weak correlations. I believe that these results should be discussed in depth to be taken with caution. 

3. In several parts of the manuscript the authors speak of a cohort study, although the data were taken from this type of study, in reality the data analyzed are cross-sectional. 

4. In the discussion section, I suggest that the authors add the limitations of their work.

5. In section 3.2, the authors repeat a lot of information that is shown in Table 3. I suggest avoiding duplication of information to make the manuscript easier to read.

6. In Table 1, the authors use asterisks to mark some notes. I suggest using other signs so as not to confuse the reader with the statistical differences found.

7. Revise the references. For example, number 2 is incomplete and number 19 has a different format.